# Theoretical Study of Vibrational Properties of Peptides: Force Fields in Comparison and Ab Initio Investigation

Nicole Luchetti [1,*] and Velia Minicozzi [2,*]

1 Department of Engineering, Campus Bio-Medico University of Rome, Via Álvaro del Portillo 21, 00128 Rome, Italy
2 Department of Physics, University of Rome Tor Vergata, and INFN, Via della Ricerca Scientifica 1, 00133 Rome, Italy
* Correspondence: n.luchetti@unicampus.it (N.L.); velia.minicozzi@roma2.infn.it (V.M.)

**Abstract:** Infrared (IR) spectroscopy is a valuable tool to obtain information about protein secondary structure. The far-infrared (FIR) spectrum is characterized by a complex combination of different molecular contributions which, for small molecules, may be interpreted with the help of quantum-mechanical (QM) calculations. Unfortunately, the high computational cost of QM calculations makes them inapplicable to larger molecules, such as proteins and peptides. In this work, we present a theoretical study on the secondary structure, molecular properties, and vibrational spectra of different peptides, using both a classical and a QM approach. Our results show that the amide I main peak value, and related quantities, such as dipole strength (DS) and transition dipole moment (TDM), depends on protein secondary structure; in particular, from QM calculations arises that $\alpha$-rich molecular systems present lower intensities than $\beta$-rich ones. Furthermore, it is possible to decouple and identify the intensity of the different contributions of the inter- and intra-molecular motions which characterize the FIR spectrum, starting from the results obtained with QM calculations.

**Keywords:** infrared; secondary structure; force field; density functional theory; molecular vibrations





## 1. Introduction

The main application of infrared (IR) and Fourier transform infrared (FTIR) spectroscopy in biophysics is to provide information about protein secondary structure [1–9] from the analysis of the amide I and amide II bands of the spectrum. The former is mainly characterized by the stretching vibrations of the C=O bonds within the protein backbone, while the latter is characterized by the in-plane stretching vibration of the C-N bonds in combination with N-H bending. Generally, the secondary structure is inferred upon deconvolution of the amide I band's different contributions and by the evaluation of the relative intensity of its components. The principal types of secondary structures, i.e., $\alpha$-helix and antiparallel $\beta$-sheets, provide absorption features localized about ten cm$^{-1}$ apart from each other.

The identification of protein secondary structure and the analysis of secondary structure changes may help unveil the mechanisms underlying protein misfolding and aggregation, one of the most important (and unsolved) issues in structural biophysics, involved in a wide variety of diseases, known as amyloidoses, and in all neurodegenerative diseases. The aggregation and fibril formation process is preceded by a conformational modification in which the protein loses its native structure which is locally replaced by a $\beta$-sheet structure, prone to the formation of aggregates. Hence, the study of the vibrational absorption spectrum can provide important information about protein fibrils' formation [10–12].

Currently, the interpretation of the experimental results is still limited by the difficulty of having a complete theoretical description. In the last decades, several theoretical approaches [13–22] have been proposed and widely used to reconstruct the IR spectra of peptides and proteins. From the theoretical point of view, one can approach the problem by

carrying out a classical molecular dynamics simulation, calculating the Fourier transform (FT) of the dipole autocorrelation function [23–26] or performing normal modes (NM) analysis calculations by diagonalizing the Hessian matrix of the total energy at the equilibrium configuration (obtained by minimizing the system) and then extracting its eigenvalues (representing the eigenfrequencies related to the IR active modes of the system). The individual C=O oscillators are coupled electrostatically; this coupling is often described by the transition dipole coupling (TDC) approximation [27–29], which delocalizes the local modes over nearby peptides in the form of Frenkel excitons [30,31]. The TDC intensity between nearby peptides varies about 10–20 cm$^{-1}$ [32] depending on several factors, such as the secondary structure length and arrangement and the intra- and inter-strand interaction potential. Unfortunately, the TDC is known not to reproduce nearest neighbor couplings, which need to be obtained from quantum mechanical (QM) calculations on small peptides [33–35].

In the present work, we perform both classical and QM calculations to compute the IR spectra of small peptides in $\alpha$-helix or $\beta$-sheet structures. The classical calculations give us the possibility of evaluating the importance of the choice of the force field (FF, the model for the interactions of all the components of a system) for the computation of the vibrational spectrum [36–39] using NM analysis. The QM calculations, based on density functional theory (DFT) [40,41], allow us to compute the transition dipole moment (TDM) and the dipole strength (DS) of the C=O oscillators in $\alpha$ and $\beta$ structures. Both TDM and DS cannot be calculated by using a classical approach in which it is not possible to consider the correlation between coupled oscillators. Moreover, QM calculations give quantitative information on the percentage of the contributions of intra-molecular vibrations, rotations, and translations [42,43], which are essential for characterizing each molecular mode present in the IR spectrum. This approach is useful for studying the far-infrared (FIR) region because of the complex combination of the different molecular motions involved. Hence, it would be particularly useful for the analysis of free-electron lasers (FEL) experiments, which, moreover, would allow for obtaining deeper structural information on biomolecules by giving the possibility of exploring their low-frequency vibrational motions and performing a structural analysis of biological materials [44–50].

## 2. Materials and Methods

The classical equations of motion for the nuclei, at the equilibrium configuration, needed to be solved to obtain the vibrational information, both in the classical and in the QM calculations.

The eigenvectors and eigenvalues were calculated by diagonalizing the Hamiltonian Hessian matrix, in the harmonic approximation.

Since the diagonalization of the energy Hessian matrix of a system with many degrees of freedom has a very high computational cost, all the systems were simulated in the gas phase, to improve computational performance.

Classical calculations were performed using GROMACS [51] v. 2020.3, while ab initio calculations were performed using Quantum ESPRESSO (QE) [52] v. 6.6 software.

### 2.1. Classical Calculations

The aim of the classical calculations is twofold: first, to identify the main vibrations that characterize the amide I band for the two most common types of secondary structure, and second, to test in which way the force field (FF)—the approximation of the interaction potential—may influence the vibrational properties of the molecular systems.

The most common functional form for the interaction potential is the following:

$$V^{tot} = \frac{1}{2} \sum_{bonds} k_i^b (\vec{r}_i - \vec{r}_{0i})^2 + \frac{1}{2} \sum_{angles} k_j^a (\theta_j - \theta_{0j})^2 + \frac{1}{2} \sum_{twists} V_n (1 + cos(n\omega - \gamma))$$
$$+ \sum_{k<l=1}^{N} \left\{ 4\epsilon_{kl} \left[ \left( \frac{\sigma_{kl}}{r_{kl}} \right)^{12} - \left( \frac{\sigma_{kl}}{r_{kl}} \right)^6 \right] + \frac{q_k q_l}{4\pi\epsilon_0} \frac{1}{r_{kl}} \right\}, \tag{1}$$

where the first three terms identify the bonded contributions (among two or three atoms and two consecutive amino acids), and the two latter terms represent the non-bonded contributions (Lennard–Jones and Coulomb) to the potential. Each FF is characterized by certain values of the reference parameters and constants ($\vec{r}_{0i}$, $\theta_{0j}$, $k_i^b$, $k_j^a$, $\sigma_{ij}$, $q_k$, and so on), generally obtained from experiments or derived from ab initio calculations [37,38]. We compare the four most used FFs: AMBER03, CHARMM22*, OPLS-AA, and GROMOS96-43a1. Since the bonded terms are the main ones responsible for the vibration frequencies characterizing the amide I band, in Table 1 we report their functional form types [53].

**Table 1.** Functional forms of force fields. Harm. = harmonic; Per. = periodic; U-B = Urey–Bradley; Mult.-per. = multiple-periodic; R-B = Ryckaert–Bellemans; Cos.-based = cosine-based.

|  | Binding | Bending | Improper | Proper |
|---|---|---|---|---|
| AMBER | Harm. | Harm. | Per. | Mult.-per. |
| CHARMM | Harm. | U-B | Harm. | Mult.-per. |
| OPLS | Harm. | Harm. | Harm. | R-B |
| GROMOS | 4th grade | Cos.-based | Harm. | Per. |

As models for $\alpha$ and $\beta$ structures, we extracted two short peptides from long MD simulations of solvated Bovine Serum Albumin (BSA, PDB-id 4f5s) and Concanavalin A (ConA, PDB-id 1vln) performed in a previous work of our group [1]:

1. $\alpha$-helix peptide from BSA protein (residues: 367–396, 29 aa long);
2. $\beta$-sheet peptide from ConA protein (residues: 108–138, 30 aa long).

The two peptides were chosen since they remained in stable $\alpha$ (peptide 1) and $\beta$ (peptide 2) conformations and they had a sufficient number of amino acids in a well-defined secondary structure, but at the same time they were not so long as to affect the computational time.

In Figure 1 we report a cartoon of the two peptides.

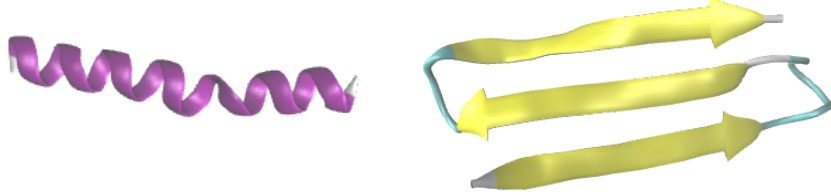

**Figure 1.** On the **left**, the $\alpha$ peptide (29 aa) and, on the **right**, the $\beta$ peptide (30 aa, right) used for classical calculations, represented in cartoon style with VMD software [54]. Legend: purple—$\alpha$-helix; yellow—$\beta$-sheet.

The peptides were minimized using three algorithms in series: steepest descent [55], conjugate gradient [56], and limited-memory Broyden–Fletcher–Goldfarb–Shanno [57]. The last one is an optimization algorithm belonging to the family of quasi-Newton methods, necessary to perform the calculation of the normal modes with GROMACS.

### 2.2. Ab Initio Calculations

We carried out DFT calculations in order to both compute quantities (such as the DS and the TDM) that cannot be estimated from a classical approach and compared the results obtained, for such quantities, by using a norm-conserving (NC) and an ultrasoft (US) BLYP [58] pseudopotential (pp). The use of pp in QE allows for the description of the pseudowave function using a plane-wave basis set.

The QM calculations were performed on two systems both constituted of a polyglycine peptide 8 aa long (58 atoms), one in $\alpha$-helix and one in antiparallel $\beta$-sheet conformations.

Both peptides were extracted from a long MD simulation in water (data not published). Glycine is the simplest amino acid, having a side chain consisting only of one hydrogen atom; hence, polyglycine peptides were chosen to have the possibility of simulating and comparing results for $\alpha$ and $\beta$ conformations at the lowest computational cost.

In Figure 2, we report a picture of the two polyglycine peptides.

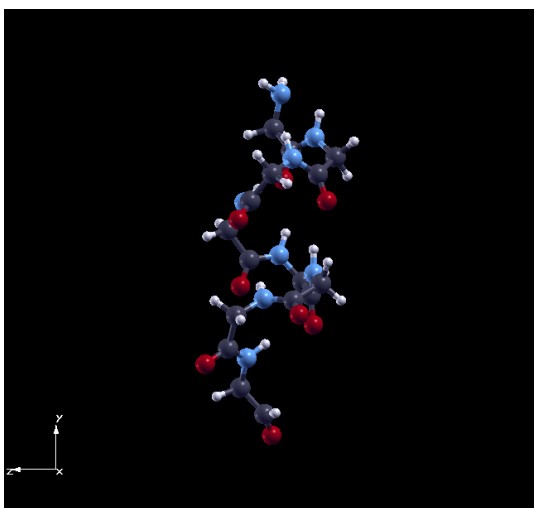 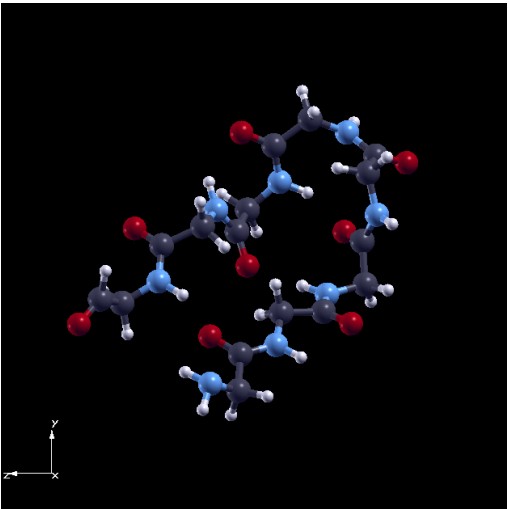

**Figure 2.** Crystal structures of $\alpha$ (**left** panel) and $\beta$ (**right** panel) polyglycine, simulated with DFT. Visualization performed with XCrysden software [59]. Legend: white—hydrogen; light blue—nitrogen; black—carbon; red—oxygen.

The simulated supercells were cubic, centered in the center of mass (COM) of the two systems, and the lattice parameters were chosen such that the distance between each edge of the simulation box and the COM was 6 Å (celldm(1)$_{\alpha,\beta}$ = 27.03, 24.27 Å). The systems were built using the Molefacture plugin of Visual Molecular Dynamics (VMD) [54] software, starting from pre-existing thermalized polyalanines and modifying their side chains.

Firstly, we performed a relaxation step to reach the minimum energy configuration for both systems. Next, using the ph.x package, we rebuilt the IR spectrum with intensities expressed in $\left(\frac{D}{\mathring{A}}\right)^2 \frac{1}{\text{amu}}$ and frequencies in cm$^{-1}$.

For each frequency mode, a self-consistent field (SCF) calculation was carried out with an accuracy threshold of $10^{-15}$ Ry. The post-processing calculations (performed with the dynmat.x package) needed to diagonalize the matrix, allowing us to obtain a file containing all the modes, which could be then visualized using the MOLDEN [60] v. 5.7 program for molecular and electronic structure processing.

After having identified the intensity and frequency of the maximum amide I peak related to the C=O stretching vibration for both $\alpha$ and $\beta$ conformations, it was possible to calculate the TDM, expressed in Debye (D), using the relationship [13,16]

$$|\Delta\mu_k| = \sqrt{\frac{h}{8\pi^2 c\nu_k}} \left|\frac{\partial\mu_k}{\partial Q_k}\right|, \tag{2}$$

where $\nu_k$ is the frequency of the k$^{\text{th}}$ mode, expressed in cm$^{-1}$, and $\left|\frac{\partial\mu_k}{\partial Q_k}\right|$ is the dipole derivative, expressed in $\left(\frac{D}{\mathring{A}}\right)\sqrt{\frac{1}{\text{amu}}}$.

Afterward, we calculated the DS, expressed in D$^2$, using the following formula [13]:

$$DS_k = 4.1058^2 \left|\frac{\partial\mu_k}{\partial Q_k}\right|^2 \frac{1}{\nu_k}, \tag{3}$$

where 4.1058 is the prefactor present in Equation (2).

We finally used an analytical mode-decoupling method proposed by Zhang et al. [42,43] that allows the decomposition of a normal mode into inter-molecular translation and libration and intra-molecular vibrational motions. To study the decoupling of motions which characterize the FIR spectrum of molecules, an ad hoc Fortran script was written [61], to separate the contributions to each whole system mode due to the intra-molecular vibrations and inter-molecular translations and librations.

For each mode of interest, it was possible to plot the weight (in percentage) of the three types of motion along each direction $(x, y, z)$, starting from the output file of the dynmat.x post-processing calculation, which contained the displacements along each direction of every atom composing the system, for all modes.

Once the displacement vectors for the inter-molecular translations and librations and intra-molecular vibrations were calculated, it was possible to determine the amplitudes of the three translations, the three principal librations, and the intra-molecular vibrations for the $k$th normal mode by evaluating the root-mean-square mass-weighted atomic displacement of all atoms in the molecule:

$$D_l^k = \sqrt{\frac{1}{N} \sum_{i=1}^{N} m_i (\delta_{i,V}^{k,l})^2} \ , \tag{4}$$

where the index $l$ refers to the three translations, the three principal librations, and the intra-molecular vibrations, the index $V$ refers to the three directions, and $i$ identifies the atom. The percentage $P_{k,l}$ of each component in the $k$th mode is given by:

$$P_{k,l} = \frac{D_l^k}{\sum_l D_l^k} \times 100 \tag{5}$$

## 3. Results and Discussion

### 3.1. Classical Normal Modes Analysis

In the following, we show the plots of the theoretical amide I bands computed using the NM analysis for each FF reported in Table 1, for the $\alpha$ and $\beta$ peptides listed in Section 2.1 (continuous green and dashed black lines, respectively). In all the plots, it is possible to notice that both $\alpha$ and $\beta$ peptides show an amide I band characterized by a more or less noisy peak (depending on the FF). The peak frequencies for $\alpha$ and $\beta$ structures are quite well distinguishable for AMBER03 and CHARMM22* FF (Figures 3 and 4). The former is located about 20 cm$^{-1}$ to the right with respect to the latter; this finding is in good agreement with the results of a previous work [1], in which NM analysis was performed on the entire BSA and ConA proteins. Conversely, OPLS and GROMOS96 FF (Figures 5 and 6) do not allow us to distinguish the main peaks for $\alpha$ and the $\beta$ structures. This may be due to the different modelization of the dihedral potential and the absence of cross-terms in the bending potential.

The FF which gives the best results is CHARMM22*, which takes into account not only a dihedral energy term (cMAP), able to model the changes in protein secondary structure, but also a cross-term in the angle bending energy (the Urey–Bradley potential term), which results in a more accurate description of the vibrational spectrum exactly in the amide I region.

As anticipated, in all cases, the presence of noise may indicate that the structures are not properly minimized with the classical approach, and it is, therefore, necessary to resort to a more accurate optimization phase, as illustrated in Section 3.2. Furthermore, when comparing the present results with the ones (both experimental and theoretical) obtained from our group in a previous work [1] (Figure S2 for experimental results and Figure S3 for computational results in the Supplementary Material of Ref. [1]), we may observe that noise depends on the number of amino acids forming the protein (or the peptide) and that the experimental and calculated spectra are in good agreement.

AMBER03 Force Field

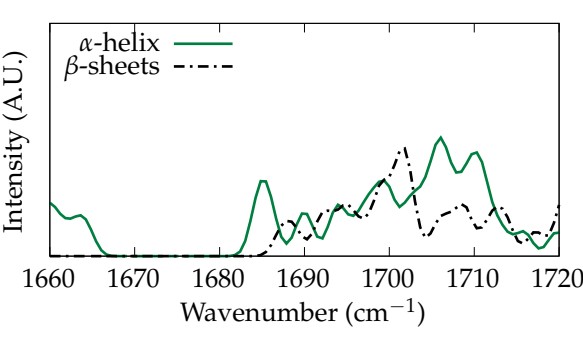

**Figure 3.** Amide I region calculated with AMBER03 FF.

CHARMM22* Force Field

**Figure 4.** Amide I region calculated with CHARMM22* FF.

OPLS-AA Force Field

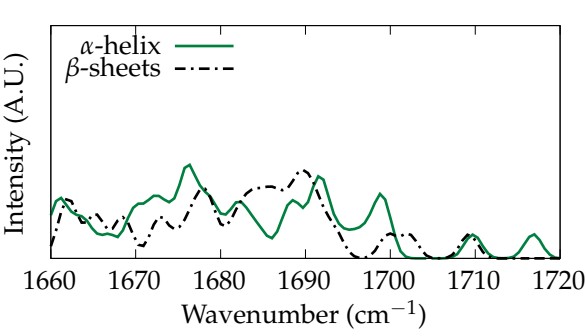

**Figure 5.** Amide I region calculated with OPLS-AA FF.

GROMOS96-4a31 Force Field

**Figure 6.** Amide I region calculated with GROMOS96-43a1 FF.

As expected with the classical approach, in addition, the intensity of the amide I peaks, as also shown in [1], strongly depends on the number of amino acids arranged in a specific secondary structure. Hence, it is important to verify whether the system preserves the $\alpha$ and $\beta$ structures during the minimization phase (and the FF), before the NM calculations.

To this end, we analyzed the Ramachandran plots of the two peptides after minimization, for each FF. In the Ramachandran plots, each type of secondary structure is well localized in a specific region of allowed values for the two torsional angles $\Phi$ (*x*-axis) and $\Psi$ (*y*-axis), in the range of $-180°$ and $+180°$. In particular, $\alpha$-helix and $\beta$-sheet conformations are located in three specific regions: $\beta$-sheet in the upper-left quadrant and $\alpha$-helix and left-handed $\alpha$-helix in the lower-left and the upper-right quadrants, respectively. Generally, in Ramachandran plots, regions' colors depend on the stability of the secondary structure (black is the most stable and white the least).

To analyze the structural geometry and reconstruct the Ramachandran plots of each simulated system, we used the Ramachandran Plot Server [62] developed by the ZLAb of UMass Chan Medical School. In Appendix A.3, the Ramachandran plots for the $\alpha$- and $\beta$-peptide for each tested FF are reported in Figures A3–A6.

CHARMM22* FF, maybe due to the cMAP term, provides the best results also regarding secondary structure determination, with all amino acids located in highly preferred regions, both for $\alpha$ and $\beta$ structures (Figure A4); amino acids in preferred regions are identified by green crosses. Conversely, GROMOS96 FF (Figure A6) provides the worst results, especially for $\beta$ structure; amino acids in less preferred regions are identified by red circled orange triangles, and amino acids in forbidden regions are identified by blue circled orange triangles. In all cases, the Ramachandran plot for the $\beta$ structure is characterized by a more inhomogeneous distribution of amino acids, probably due to the presence of loop segments necessary to obtain a $\beta$-sheet.

In Appendix A.1, the complete classical simulated IR spectra, in the range of $[0 \div 4000]$ cm$^{-1}$, are reported in Figure A1. It is possible to distinguish the amide II and the amide I bands (in the range of $[1300 \div 2000]$ cm$^{-1}$) and the amide A and B bands (in the range of $[3000 \div 3500]$ cm$^{-1}$). This last region is mainly influenced by the Fermi resonance between the N-H bending vibration characterizing the amide II band and the N-H stretching vibration.

### 3.2. Ab Initio Phonon Calculation

Ab initio calculations, unlike classical ones, allow us to see the electronic coupling among carbonyl groups disposed along the main chain. As expected, although the two polyglycine peptides had the same number of amino acids, the amide I peak intensity for the $\beta$ conformation was greater than the one for the $\alpha$ conformation; indeed, the intensity of the amide I region depends on the local secondary structure arrangement, as we can notice from Figures 7 and 8 and Table 2 and 3.

To solve Equations (2) and (3), we directly extracted from the post-processing outputs the intensity and the frequency related to the most intense peak in the MIR region of the spectrum for all the simulated peptides:

– $\alpha$ peptide with NC PP (w/ QM relaxation): 11.7 (D/Å)$^2$/amu at 1650 cm$^{-1}$;
– $\alpha$ peptide with US PP (w/ QM relaxation): 12.8 (D/Å)$^2$/amu at 1640 cm$^{-1}$;
– $\alpha$ peptide with NC PP (w/o QM relaxation): 12.6 (D/Å)$^2$/amu at 1779 cm$^{-1}$;
– $\beta$ peptide with NC PP (w/ QM relaxation): 14.9 (D/Å)$^2$/amu at 1623 cm$^{-1}$;
– $\beta$ peptide with US PP (w/ QM relaxation): 18.2 (D/Å)$^2$/amu at 1617 cm$^{-1}$;
– $\beta$ peptide with NC PP (w/o QM relaxation): 18.8 (D/Å)$^2$/amu at 1734 cm$^{-1}$.

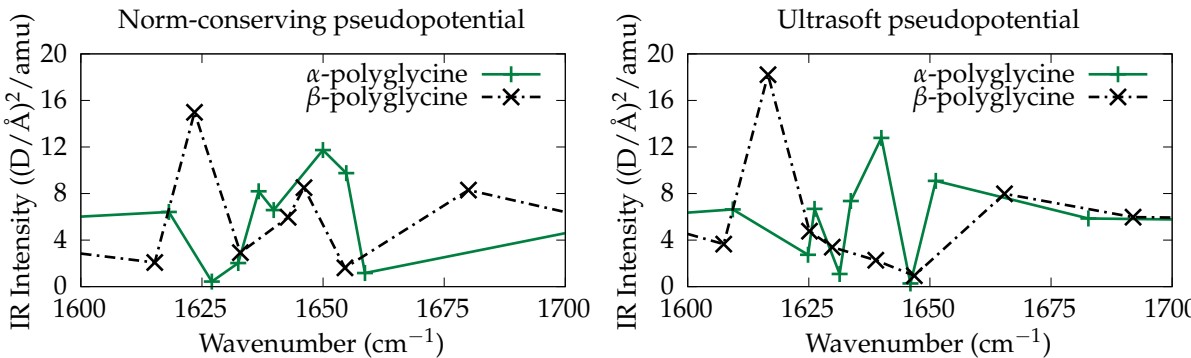

**Figure 7.** Amide I region simulated with DFT calculations.

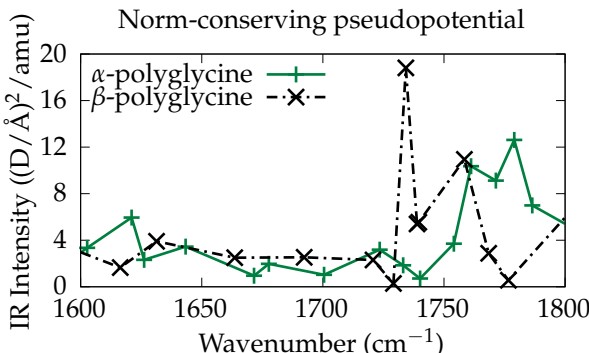

**Figure 8.** Amide I region simulated with DFT calculations, referred to Table 3.

**Table 2.** Wavenumbers and intensities of the C=O stretching vibration (NC pp / US pp).

|  | Frequency (cm$^{-1}$) | TDM (D) | DS (D$^2$) |
|---|---|---|---|
| $\alpha$-helix | 1650/1640 | 0.346/0.362 | 0.120/0.131 |
| $\beta$-sheets | 1623/1617 | 0.394/0.436 | 0.155/0.190 |

**Table 3.** Wavenumbers and intensities of the C=O stretching vibration (NC pp w/o QM relaxation).

|  | Frequency (cm$^{-1}$) | TDM (D) | DS (D$^2$) |
|---|---|---|---|
| $\alpha$-helix | 1779 | 0.346 | 0.120 |
| $\beta$-sheets | 1734 | 0.428 | 0.183 |

In the last table reported above (Table 3), we show the results for the simulated polyglycines with NC pp without the QM relaxation of atomic coordinates, only performing a classical minimization with GROMACS (using the algorithms in series cited in Section 2.1). The amide I regions referring to the table are shown in Figure 8. The two peaks are not well solved and the frequencies are all blue-shifted with respect to the ones shown in the left panel of Figure 7.

The ab initio results are qualitatively in agreement with the theoretical and experimental existing literature; the experimental evidence of the FTIR spectra of proteins shows that oscillating carbonyl groups disposed along the main chain within a local $\beta$ conformation have a stronger coupling as compared with a local $\alpha$ conformation (see Table 3). It has been observed that the DS for the $\alpha$ conformation is around 0.120 D$^2$, while it is 0.180 D$^2$ for the $\beta$ conformation [1,16,63,64], with values for TDM of 0.348 D and 0.390 for $\alpha$-polyglycine and $\beta$-polyglycine, respectively [16].

As we can see from Tables 2 and 3, the ratio $\frac{DS_\beta}{DS_\alpha}$ is:

–    1.30 (NC pp w/ QM relaxation);

- 1.45 (US pp w/ QM relaxation);
- 1.50 (NC pp w/o QM relaxation).

Values obtained for the DS intensity from the calculations without QM relaxation (Table 3 and Figure 8) are in good agreement with the experimental findings, as also argued in our previous work [1]. However, maybe due to a not proper minimization of the peptides, the peak frequencies are shifted to higher values (like in free chromophores). The results obtained for the C=O stretching related peak frequency, from the calculations performed with QM relaxation (Table 2 and Figure 7), are in very good agreement with the previous experimental and theoretical literature, while the DS intensities are underestimated for the $\beta$ conformation simulated with NC pp, and they are overestimated for the $\alpha$ conformation simulated with US pp.

As expected, in all simulated systems, the frequency of the most intense peak related to the $\alpha$ structure is located about 20 cm$^{-1}$ to the right of the peak frequency related to the antiparallel $\beta$ structure [65,66]

In Appendix A.2, the complete simulated IR spectra in the range of [0 ÷ 3500] cm$^{-1}$, obtained from ab initio calculations, are reported in Figure A2. It is possible to distinguish the amide II and the amide I bands (in the range of [1300 ÷ 1700] cm$^{-1}$) and the amide A and B bands (in the range of [3000 ÷ 3500] cm$^{-1}$). It is also possible to observe some molecular modes in the FIR region, which are not visible from classical simulations.

### 3.3. Decomposition of Molecular Modes

As described in Section 2, the Fortran script allows for the identification of the different contributions of each mode to the motions, in the three directions of the space. The analysis may be applied to the whole IR spectrum, but it is very useful to understand what happens in the FIR region which is the most complex of the IR spectrum of biological macromolecules, due to the overlapping of several molecular motions.

We decided to apply the analysis to the polyglycine peptides simulated both with NC and US pp with QM relaxation and to extract and analyze results for the first 20 molecular modes (in the frequency range [100 ÷ 300] cm$^{-1}$) obtained from the ph.x calculation. In the following plots, we report the decomposition analysis of the systems cited above. We separate the contributions along the three directions for translation and libration motions (purple—$x$ axis; orange—$y$ axis; green—$z$ axis), as proposed by Zhang et al. [42,43].

As we can observe from the following plots, major contributions arise from librations and intra-molecular vibrations. The tiny intensities corresponding to the translation motions for all the systems may be due to the fact that the systems are composed of a single molecular unit in the cell, so we cannot see translations among more molecules in the cell.

It is interesting to notice that while the libration motions intensities for $\alpha$ polyglycine seem to depend on the pp used in the calculations (Figures 9 and 10), the ones for $\beta$ polyglycine do not (Figures 11 and 12); for both pp, we obtain $y$ and $z$ as preferred directions for the libration motions.

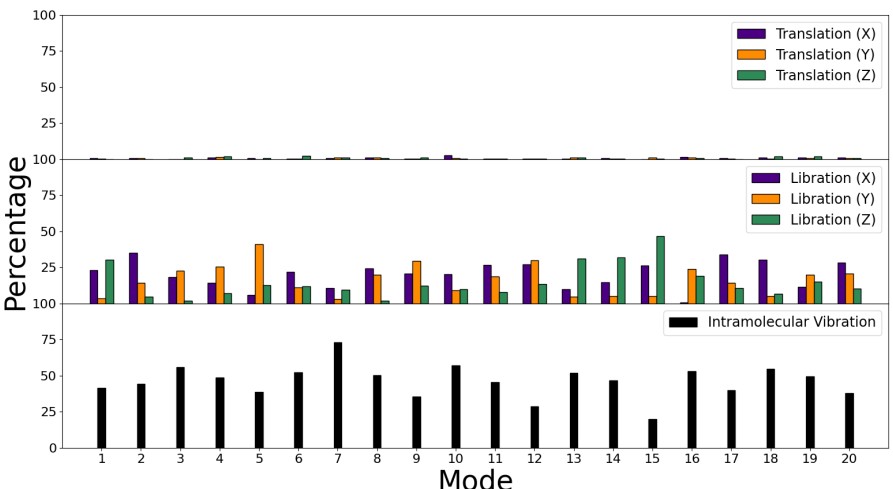

**Figure 9.** Decomposition of molecular modes for *α*-polyglycine with NC pp.

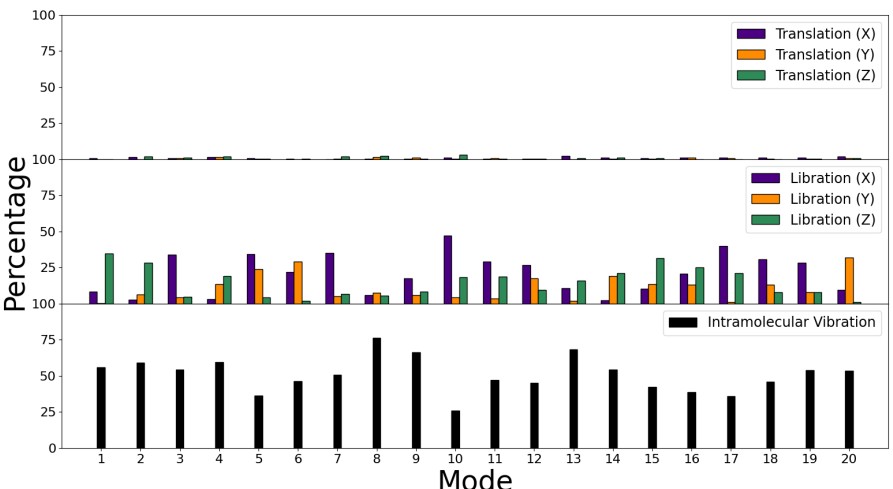

**Figure 10.** Decomposition of molecular modes for *α*-polyglycine with US pp.

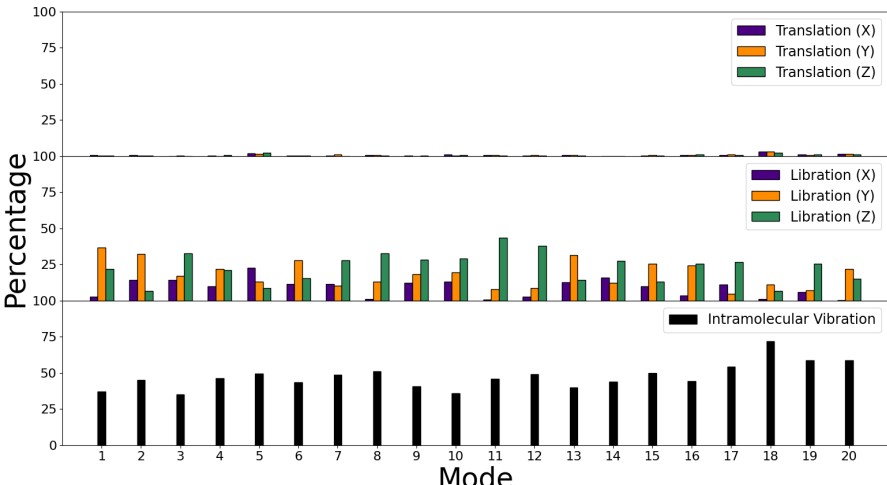

**Figure 11.** Decomposition of molecular modes for *β*-polyglycine with NC pp.

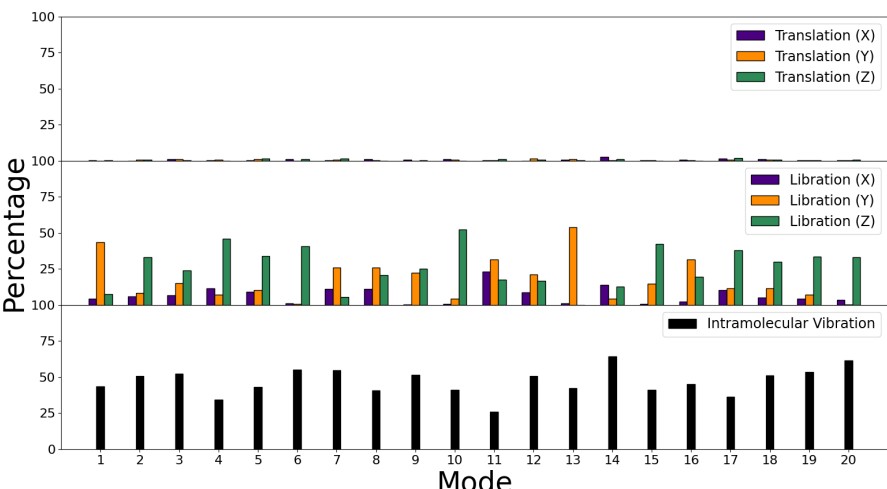

**Figure 12.** Decomposition of molecular modes for $\beta$-polyglycine with US pp.

## 4. Conclusions

The present study aims to accurately determine the main spectral features arising from the organization of protein secondary structure by comparing theoretical results obtained with different approaches. The amplitude of the DS provides important information about the interaction among carbonyl groups at the microscopic level since it is involved in models for local and extended vibrational excitation. Our findings confirm that the amplitude of the dipole strength in the C=O groups depends on protein secondary structure, according to the previous literature.

Although classical calculations are not able to reproduce the correlations among different carbonyl groups disposed of along the protein backbone, they are very promising. Classical calculations can distinguish the different contributions of the C=O stretching vibration to the spectrum due to the protein local arrangement in $\alpha$-helix or $\beta$-sheet secondary structures. To summarize, we found out that peptides/proteins in the $\beta$ conformation present an amide I peak located at lower frequencies compared to the one characterizing the $\alpha$ conformation.

Furthermore, the comparative study of the four different FFs allows us to conclude that the parameters and the functional form of the interaction potential provided by CHARMM22$^*$ FF make it the best suited to perform studies on the vibrational properties of biological molecules.

Since electronic contributions are lacking in the classical approach, we performed DFT calculations on two polyglycine peptides in $\alpha$ and $\beta$ conformations. The ab initio results confirm classical ones and give a more reliable estimate of the TDM and the DS, reproducing results that are in very good agreement with the existing experimental and theoretical ones. We also demonstrated that a quantum-chemical relaxation of the starting structures is necessary to obtain better spectral features, both in classical and ab initio calculations.

The decomposition of inter- and intra-vibrational motions provides a useful tool that allows the users to study the characterization of each molecular mode by analyzing the intensity of the three principal contributions (translations, librations, and vibrations) along the different directions. The present study about biomolecules could easily be extended to the analysis of the vibrational properties of inorganic compounds.

**Author Contributions:** Conceptualization, N.L. and V.M.; investigation, N.L.; writing—original draft preparation, N.L. and V.M.; writing—review and editing, N.L. and V.M.; visualization, N.L.; supervision, V.M. All authors have read and agreed to the published version of the manuscript.

**Funding:** This research received no external funding.

**Institutional Review Board Statement:** Not applicable.

**Informed Consent Statement:** Not applicable.

**Data Availability Statement:** Not applicable.

**Acknowledgments:** N.L. and V.M. acknowledge the Super-Computing Inter-university Consortium CINECA for support and high-performance computing resources under the Italian Super-Computing Resource Allocation (ISCRA) initiative. The authors acknowledge the support of the Italian Institute of Nuclear Physics (INFN).

**Conflicts of Interest:** The authors declare no conflict of interest.

**Sample Availability:** The structures of simulated systems are available from the authors.

**Abbreviations**

The following abbreviations are used in this manuscript:

FIR     far-infrared
MIR     mid-infrared
QM      quantum-mechanical
TDC     transition dipole coupling
TDM     transition dipole moment
DS      dipole strength
FF      force field
aa      amino acid/s
NM      normal mode/s
DFT     density functional theory
pp      pseudopotential
MD      Molecular Dynamics
VMD     Visual Molecular Dynamics
NC      norm-conserving
US      ultrasoft

**Appendix A**

*Appendix A.1*

Complete classical FIR and MIR spectra.

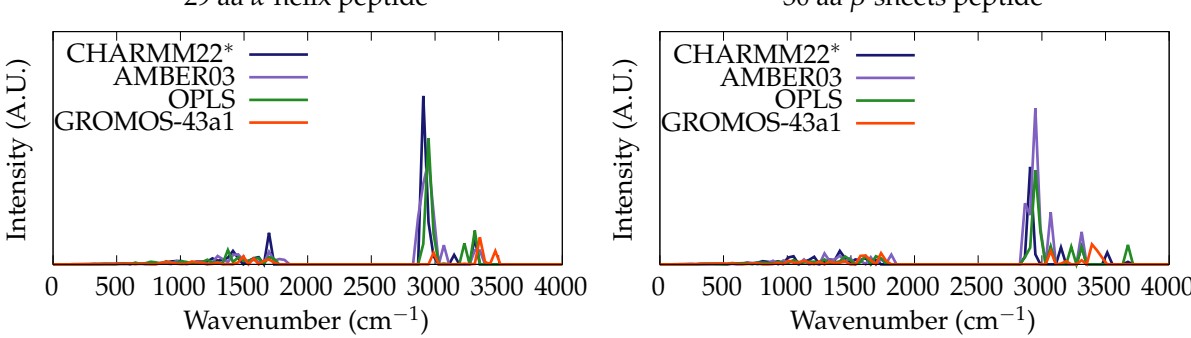

**Figure A1.** Total IR spectra for $\alpha$ and $\beta$ peptides, simulated with classical calculations.

*Appendix A.2*

Complete ab initio FIR and MIR spectra.

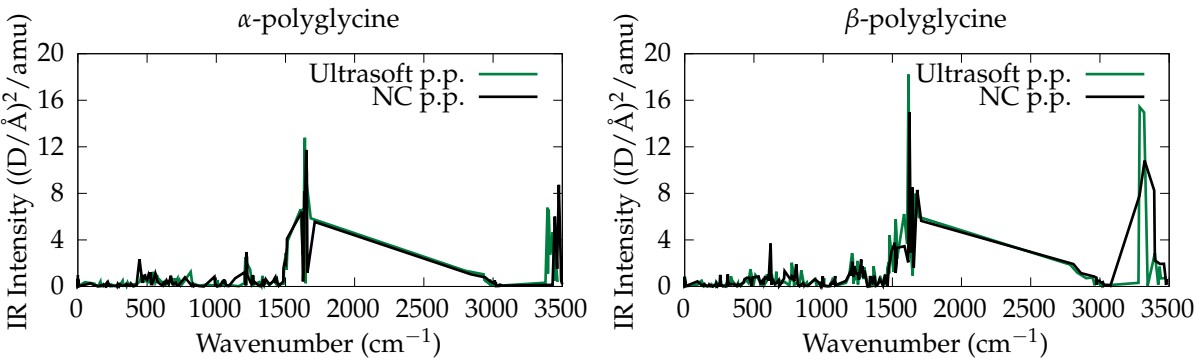

**Figure A2.** Total IR spectra for $\alpha$ and $\beta$ polyglycine, simulated with DFT calculations.

*Appendix A.3*

Ramachandran plots for $\alpha$ and $\beta$ peptides.

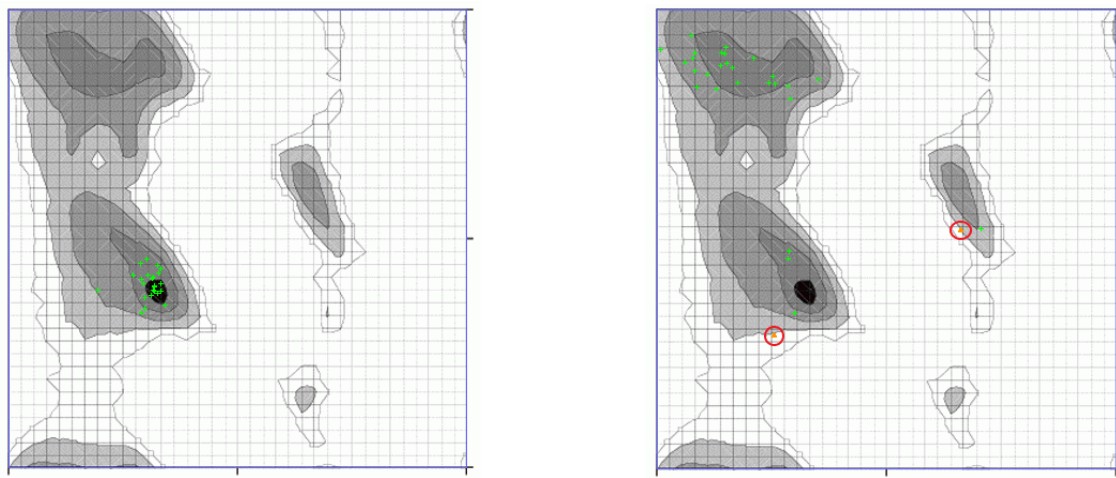

**Figure A3.** Ramachandran plots with AMBER03 FF for $\alpha$ (**left**) and $\beta$ (**right**) structures.

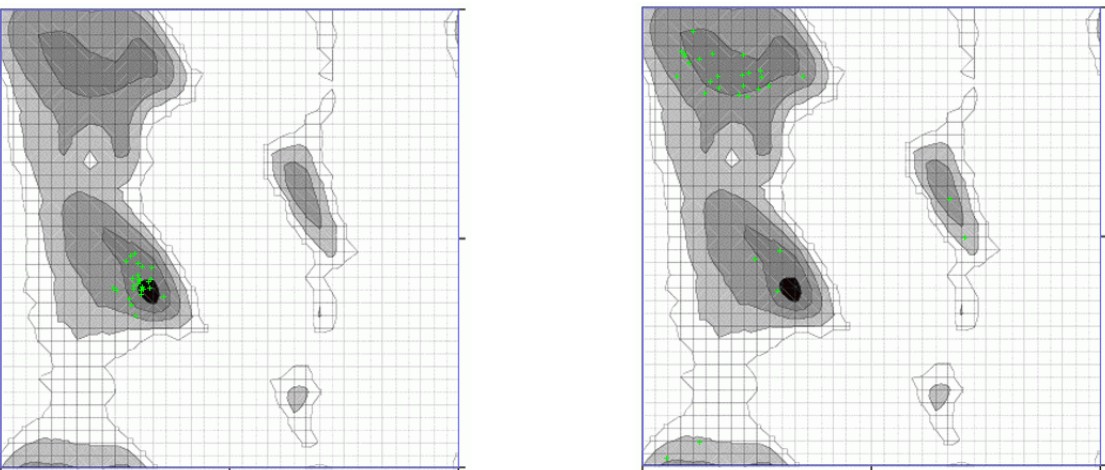

**Figure A4.** Ramachandran plots with CHARMM22* FF for $\alpha$ (**left**) and $\beta$ (**right**) structures.

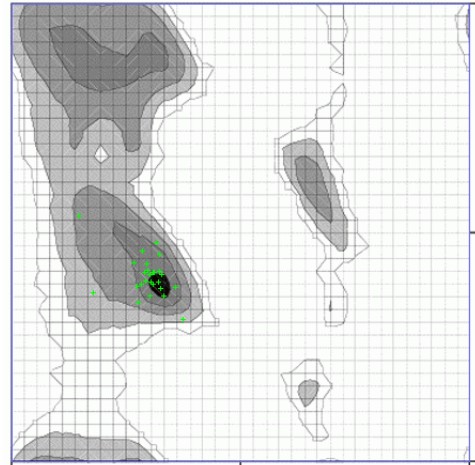 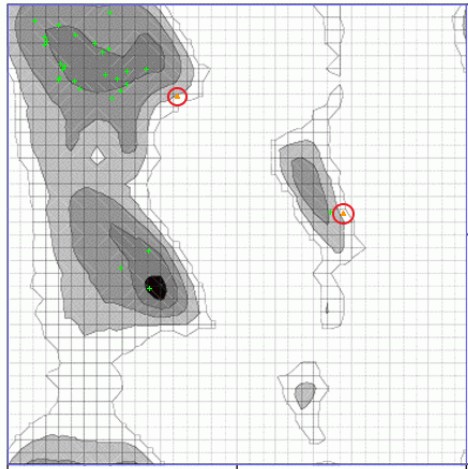

**Figure A5.** Ramachandran plots with OPLS FF for *α* (**left**) and *β* (**right**) structures.

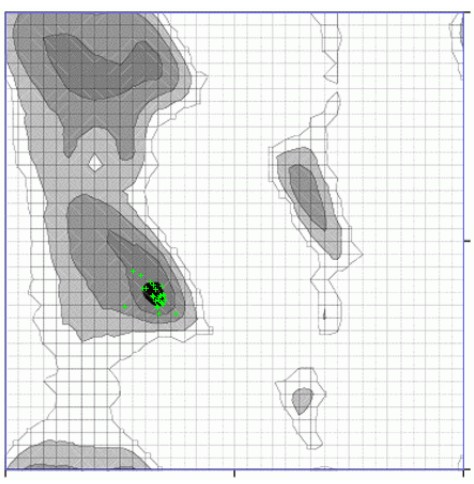 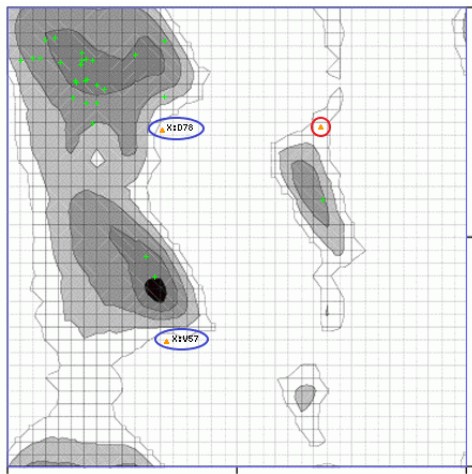

**Figure A6.** Ramachandran plots with GROMOS96 FF for *α* (**left**) and *β* (**right**) structures.

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
