# Peer review of "Theoretical Study of Vibrational Properties of Peptides: Force Fields in Comparison and Ab Initio Investigation"

_condensedmatter, doi:10.3390/condmat7030053_

Round 1

Reviewer 1 Report

This MS has the ambitious goal of reproducing the IR signal of proteins and peptides, in particular of the amide I region, which notoriously shows different shape of the corresponding peak, depending on the secondary protein structure.

Despite the important goal, the authors imply a set of several approximation, that need to be evaluated properly. The calculations that they call “quantum mechanics” is performed in harmonic approximation, while the classical ones are performed by taking into account the anharmonicity but using a force-field. Comparing ones with the others is like to compare apples with pears. Thus, in my opinion, this MS suffers from several flaws that should be fixed or somehow put into a context before publication.

I report my main concerns below.

1) The author choose to use Quantum Espresso (QE) as their ab initio reference for their gas phase calculations. This is a very weird choice, since QE is designed for periodic systems and it employs a plane wave basis set. There are many other ab initio codes with gaussian basis set that could have been employed and that are more accurate. In addition, the QE DFT functionals are less accurate than the traditional B3LYP for gas phase calculations.

2) The authors chose to not include the solvent in their calculations. This is quite a strong limitation, since their final goal is to compare with the experiments.

3) If the gas phase is the only choice for ab initio harmonic frequency calculations, GROMACS, CHARM and AMBER force-fields have been designed condensed phase, i.e. for aqueous or liquid water solutions.

It does not make sense to apply these force-field to gas phase molecules.

4) The force-fields considered by the authors do not include polarization effects, which are important for reproducing vibrational frequencies of biomolecules (see for example Gabas et al. JCTC 16(6), 3476-3485 (2020)). I suggest the authors to use the AMOEBABIO18 force field available with TINKER suite fo codes.

5) The comparison between the classical and quantum mechanical values should be performed at the same level of approximation, I.e. either harmonic or anharmonic.

Finally, I have one minor issue: I can not understand why there are so many translation modes instead of only three. Are these frustated translations?

Reviewer 2 Report

The paper by Luchetti and Minnicozzi is a theoretical work on secondary structure, molecular properties, and vibrational spectrum of different peptides, and compares classical and QM approaches. Their conclusions support the evidence that the amide I main peak, dipole strength, and transition dipole moment depend on the protein secondary structure. QM calculations prove that α-rich molecular systems present lower intensities than ß-rich ones. They separate and identify the intensity of the different contributions of inter- and intra-molecular motions in the FIR spectrum, based on the results of QM calculations.

The work is rigorous and original. In particular, as the author stated in the introduction, these results could be useful in the analysis of next FEL experiments, because they would allow to obtain deeper structural information on biomolecules by correctly exploring low frequency vibrational motions.

There are just some minor points that I suggest to modify:

·        Table 1. In the caption there is the explanation of the titles of the Table. However, in one side there is “Cos.” and in another “Cos.-based”. They should be the same.

·        Page 3, line 93. The author should explain why they chose these parts of those proteins, if there is a reason. Otherwise, they should write if they tried to do the same with other peptides.

·        Fig. 2: the axis in white with grey background are really difficult to appreciate.

·        Figures 3-6. I suggest to use continuous and dashed lines in order to help the authors printing in B&W.

·        Table 3. Are error bars present?

·        Figures 9 and 10: legends are too small to be read.

Reviewer 3 Report

This paper works on the theoretical study of far and mid-IR spectra of peptides, comparing classical force fields results and ab-initio computations. The work focuses on typical alpha-helix and beta-sheets with reasonable length. IR spectra computed in the mid-IR region (mainly the amide-I mode) and far-IR region, are given. Transition dipole moment, IR intensity, mode decomposition, are presented. Even though I'm happy to see more people are interested in working on peptides and force fields and IR spectroscopy, I cannot recommend the publication of this work. There are too many missing pieces in this work, showing that the authors need to really think what they try to say in a paper.

Just give a few examples, to prove my point.

1. Figures 3 and 4, which are the main results. Are they computed IR spectra (not easily tell by figure caption)? If so, they should be compared with experimental results and discussed. If not, what are they, why they are many peaks in the amide-I region. Take Figure 3, for example, the basic feature of the amide-I spectra for alpha-helices and beta-sheets are well documented in literature. Noise in the computed spectra is the least concern here.

2. The title includes the concept of "far IR), however, the far IR results are in the appendix and not are not discussed. It is interesting to work on the far IR signature of peptides, but the manuscript didnot do a reasonable job. In introduction, Abstract, Results, discussion, Conclusion, far-IR content is missing somewhere far away.

3. The statement in the Abstract is not fully supported. For example, "...that an antiparallel β-sheet environment is more prone to delocalize the on-site C=O stretching vibration through coupling mechanisms between carbonyl groups" is stated in the Abstract, however, the word "delocalize" are totally mentioned twice, one in Abstract, one in introduction. I look for in-depth discussion regarding the results obtained from supercells of peptides, but did not find.

Round 2

Reviewer 1 Report

The authors properly replied to all my questions.

Author Response

Dear Reviewier,

thank you very much for your prevoius comments and suggestions, which helped us improving the manuscript.

Reviewer 2 Report

The authors modified and improved the manuscript. I think it can be published as it is.

Author Response

(The authors gave the same response as above.)

Reviewer 3 Report

In the previous review of this reviewer,  just a few issues in this manuscript were pointed out, which was said very clear in the report. The author did not reasonably address a single one, rather, they simply changed the title, which I found is highly unacceptable.

The overall structure of the manuscript was and is flawed, the revision did not solve the issue.

Author Response

Dear Reviewer,

in the new response we tried to address more carefully your comments.
